# Predicting Malignant Lymph Nodes Using a Novel Scoring System Based on Multi-Endobronchial Ultrasound Features

**DOI:** 10.3390/cancers14215355

**Published:** 2022-10-30

**Authors:** Momoko Morishita, Keigo Uchimura, Hideaki Furuse, Tatsuya Imabayashi, Takaaki Tsuchida, Yuji Matsumoto

**Affiliations:** 1Department of Endoscopy, Respiratory Endoscopy Division, National Cancer Center Hospital, 5-1-1 Tsukiji, Chuo-ku, Tokyo 104-0045, Japan; 2Department of Thoracic Oncology, National Cancer Center Hospital, 5-1-1 Tsukiji, Chuo-ku, Tokyo 104-0045, Japan; 3Department of Respiratory Medicine, National Center for Global Health and Medicine, 1-21-1 Toyama, Shinjuku-ku, Tokyo 162-8655, Japan

**Keywords:** bronchoscopy, cytology, endobronchial ultrasound, transbronchial needle aspiration, histology, lung cancer, elastography

## Abstract

**Simple Summary:**

Endobronchial ultrasound (EBUS) features help differentiate between benign and malignant lymph nodes (MLNs) during transbronchial needle aspiration (TBNA). B-, power/color Doppler, and elastography modes are used during EBUS-TBNA. However, only few studies have assessed them simultaneously. This study evaluated multi-EBUS features (B-, power/color Doppler, and elastography modes) and established a novel scoring system. Multivariable analysis revealed that short axis (>1 cm), heterogeneous echogenicity, absence of central hilar structure, presence of coagulation necrosis sign, and blue-dominant elastographic images were independent predictors of MLNs. At three or more EBUS features predicting MLNs, our scoring system had high sensitivity and specificity (77.9 and 91.8%, respectively). The area under the receiver operating curve (AUC) was 0.894 (95% confidence interval (CI): 0.868–0.920), higher than that of B-mode features alone (AUC: 0.840 (95% CI: 0.807–0.873)). Our novel scoring system could predict MLNs more accurately than B-mode features alone.

**Abstract:**

Endobronchial ultrasound (EBUS) features with B-, power/color Doppler, and elastography modes help differentiate between benign and malignant lymph nodes (MLNs) during transbronchial needle aspiration (TBNA); however, only few studies have assessed them simultaneously. We evaluated the diagnostic accuracy of each EBUS feature and aimed to establish a scoring system to predict MLNs. EBUS features of consecutive patients and final diagnosis per lymph node (LN) were examined retrospectively. In total, 594 LNs from 301 patients were analyzed. Univariable analyses revealed that EBUS features, except for round shape, could differentiate MLNs from benign LNs. Multivariable analysis revealed that short axis (>1 cm), heterogeneous echogenicity, absence of central hilar structure, presence of coagulation necrosis sign, and blue-dominant elastographic images were independent predictors of MLNs. At three or more EBUS features predicting MLNs, our scoring system had high sensitivity (77.9%) and specificity (91.8%). The area under the receiver operating curve (AUC) was 0.894 (95% confidence interval (CI): 0.868–0.920), which was higher than that of B-mode features alone (AUC: 0.840 (95% CI: 0.807–0.873)). The novel scoring system could predict MLNs more accurately than B-mode features alone. Multi-EBUS features may increase EBUS-TBNA efficiency for LN evaluation.

## 1. Introduction

Endobronchial ultrasound-guided transbronchial needle aspiration (EBUS-TBNA) is widely used and recommended by various worldwide guidelines for diagnosing hilar and mediastinal lesions and staging lung cancer [1,2,3]. Two meta-analyses have reported that the cumulative sensitivity and specificity of EBUS-TBNA in non-small cell lung cancer (NSCLC) staging were 88–93% and 100%, respectively [4,5]. The diagnostic accuracy of EBUS-TBNA is also reported to be higher than that of computed tomography (CT) and positron emission tomography (PET) [1,6]. Recently, a cohort study reported that appropriate staging among patients with NSCLC preoperatively contributed to survival benefits [7]. Another retrospective study also showed that N staging with EBUS-TBNA was significantly correlated with the long-term survival of patients with NSCLC [8]. In patients with NSCLC, EBUS-TBNA specimens are used to simultaneously search for targeted driver gene mutations using next-generation sequencing [9,10]. Therefore, tumor tissue sampling using EBUS-TBNA is exceedingly important in lung cancer treatment.

Evaluating EBUS features obtained during EBUS-TBNA enables efficient procedures, especially in sedated EBUS-TBNA, when the available time is limited. Various studies conducted using B-mode features alone have reported the effectiveness of each EBUS feature and prediction score of malignant lymph nodes (MLNs) [11,12,13,14]. Interestingly, a prospective study using artificial intelligence evaluating only EBUS B-mode features (Canada lymph node (LN) score [11] (Figure 1)) showed that the accuracy of predicting MLNs was 72.9% [15]. Therefore, a scoring system that can predict MLNs during EBUS-TBNA more accurately is required. A retrospective study of 1,061 LNs classified B-mode features into six categories (short axis, shape, margin, echogenicity, absence of central hilar structure (CHS), and presence of coagulation necrosis sign (CNS)), which are widely used for differentiating between benign LNs and MLNs (Figure 1) [16]. Vascular patterns in power/color Doppler mode and elasticity in elastography mode have also been reported to be useful in such differentiation [17,18]. Power/color Doppler mode assesses vascular patterns within LNs and classifies them into grades 0–3, suggesting MLNs in grades 2–3, wherein the blood flow is relatively abundant (Figure 1) [17]. Elastography, an ultrasound technique, visualizes the relative strain of tissue compressibility within a single direction; EBUS elastography enables the detection and color-coding of the strain of LNs [18,19,20,21]. The stiff, intermediate, and soft tissues are shown in blue (suggestive of MLNs), green, and yellow or red (suggestive of benign LNs), respectively (Figure 1). A recent meta-analysis showed that EBUS elastography can predict MLNs with high sensitivity and specificity [18]; however, to the best of our knowledge, no study to date has combined all multi-EBUS features (B-, power/color Doppler, and elastography modes) and established a prediction model for MLNs. Therefore, we hypothesized that combining them would help predict MLNs more accurately.

This study aimed to evaluate the diagnostic accuracy of each EBUS feature using prospectively recorded registry data and to establish the best scoring system for predicting MLNs.

## 2. Materials and Methods

### 2.1. Patients

Data from consecutive patients who underwent EBUS-TBNA for diagnosis or staging of lung cancer at the National Cancer Center Hospital, Tokyo, Japan, between January 2019 and December 2019 were retrospectively collected from our prospective observational registry. All EBUS features were acquired and assessed during EBUS-TBNA by two or more bronchoscopists with over 10 years of bronchoscopic experience. They evaluated and classified LNs before obtaining the results of rapid on-site evaluations, so as to not be influenced by the pathological results. To evaluate the diagnostic accuracy of each EBUS feature, the final diagnosis per LN was analyzed.

This study was approved by the Institutional Review Board of the National Cancer Center Hospital, Tokyo, Japan, (No. 2018-354) and registered at the Japan Registry of Clinical Trials (jRCT) (registration number: jRCT1032190168). The requirement for written informed consent was waived due to the retrospective observational nature of the study.

### 2.2. EBUS-TBNA Procedure and Pathological Diagnosis

A convex probe ultrasound bronchoscope (BF-UC260FW or BF-UC290F; Olympus, Tokyo, Japan) was used with intravenous midazolam or propofol; lidocaine spray was used as the local anesthetic. Several studies have indicated that there is slight change in diagnostic yield according to the gauge of the needle [22,23]; therefore, a 22- or 25-gauge needle was selected by the operator.

We used a dedicated ultrasound processor (EU-ME2 PREMIER; Olympus) to observe the B-mode features, vascular patterns with the power/color Doppler mode, and elasticity with the elastography mode at each LN’s maximum diameter. We punctured the LN according to the EBUS features, its size on CT, and the maximum standardized uptake value of PET-CT. Punctures were performed 2–4 times on each LN; each puncture consisted of approximately 20–30 strokes with negative pressure.

The final diagnosis of each LN was confirmed by pathological findings obtained using EBUS-TBNA or surgery. If both EBUS-TBNA and surgical specimens were obtained, the diagnosis obtained using the latter was preferred. In cases where only EBUS-TBNA specimens were obtained and diagnosed as benign, follow-up details using CT more than 6 months later were referred to confirm the absence of disease progression. Rapid on-site cytological evaluation was performed for all patients.

### 2.3. EBUS Image Categories

B-mode features were evaluated using the following six categories (Figure 1) [16]: short axis (>1 cm or <1 cm), shape (round or oval), echogenicity (heterogeneous or homogeneous), margin (distinct or indistinct), CHS (presence or absence), and CNS (presence or absence). When the long axis was 1.5 times longer than the short axis, it was defined as an oval shape. When more than half of the margin was visible, it was defined as a distinct margin. When a flat, linear, and hyperechoic region was found in the center of the LN, CHS was considered present. When a hypoechoic area without blood flow was detected in the LN, the CNS was considered present.

Vascular patterns were evaluated using the power/color Doppler mode and classified into grades 0–3 (Figure 1) [12,17]. Grade 0 represents the smallest amounts of blood flow, grade 1 represents a few straight vessels running to the center of the LN, grade 2 represents a few punctate vessels, and grade 3 represents aberrantly developed vessels. Previous studies have reported that grades 0–1 suggest benignancy, whereas grades 2–3 suggest malignancy [12,17]; therefore, we divided grades 0–1 and 2–3 as binary variables.

EBUS elastographic images were qualitatively classified into blue, intermediate (partially blue), and non-blue (green, yellow, and red), according to the dominant color of LNs [19]. We divided EBUS elastography images into blue and others (including intermediate and non-blue) (Figure 1) as binary variables as well as B-mode features and vascular patterns. Ultrasound elastography is classified into two types: strain elastography and shear wave elastography. In this study, we used strain elastography, which can be used for EBUS.

Each EBUS feature per LN was compared with the final diagnosis to evaluate the predictive accuracy of MLNs.

### 2.4. Statistical Analyses

Descriptive statistics are presented as frequencies with percentages or medians (ranges). The sensitivity, specificity, positive predictive value, negative predictive value, and accuracy of each EBUS feature were calculated per lesion using standard definitions. Fisher’s exact test was used to compare the two groups. All EBUS features potentially predictive of MLNs were analyzed using multivariable logistic regression. Using the EBUS features selected by multivariable logistic regression, we summed the number of features and established a novel scoring system. The receiver operating characteristic (ROC) curve analysis and the Youden index were used to develop the predictive scoring system of MLNs. Delong’s test was used to compare the two ROC curves [24]. *p*-values < 0.05 were considered statistically significant. The EZR software version 1.29 was used for statistical analyses (Saitama Medical Center, Jichi Medical University, Saitama, Japan) [25].

## 3. Results

### 3.1. Patients and Lymph Nodes

During the study period, EBUS-TBNA was performed in 345 patients (677 lesions). Among them, 302 patients (597 LNs) underwent EBUS-TBNA for the diagnosis of LNs. One patient (three LNs) was excluded because of missing clinical data. Thus, 594 LNs from 301 patients were analyzed. The flow chart of patient and lymph node recruitment is shown in Figure 2.

The patient characteristics are summarized in Table 1. The patients included 199 men (66.1%), with a median age of 66 years. Past or current smokers constituted 227 (75.4%) patients; 271 patients (90.0%) had malignant tumors and 233 patients (77.4%) had lung cancer. There was only minor bleeding in 21 cases (7.0%) without any significant complications. The median number of target LNs per patient was 2 (range: 1–7), while the median number of EBUS-TBNA punctures per patient was 5 (range: 2–16).

The characteristics of LNs are summarized in Table 2. There were 177 (29.8%) lower paratracheal LNs and 162 (27.3%) subcarinal LNs. Chest CT showed 327 LNs (55.1%) with a short axis >1 cm, and the median short axis was 1.1 cm. PET results were available for 486 LNs (81.8%), among which 359 LNs (60.4%) had 18F-fluorodeoxyglucose uptake values >2.5. The final diagnosis was benign for 219 LNs (36.9%).

### 3.2. Diagnostic Yields of Each EBUS Feature for MLNs

Table 3 summarizes the number of MLNs for each EBUS feature and the diagnostic yields of each EBUS feature for MLNs. Regarding B-mode features, 319 LNs (53.7%) had a short axis >1 cm, 133 LNs (22.4%) were a round shape, 509 LNs (85.7%) had distinct margin, 399 LNs (67.2%) had heterogeneous echogenicity, 438 LNs (73.7%) had absence of CHS, and 107 LNs (18.0%) had the presence of CNS. Regarding vascular patterns with power/color Doppler mode, 217 (36.5%), 117 (19.7%), 171 (29.3%), and 86 LNs (14.8%) were of grade 0, 1, 2, and 3, respectively. Regarding EBUS elastographic images, 163 LNs (27.4%) were blue, 268 LNs (45.1%) were intermediate, and 163 LNs (27.4%) were non-blue.

In univariable analyses, all EBUS features, except shape, were associated with the prediction of MLNs. In the multivariable analysis, all EBUS features revealed that the following five features were independent predictive factors of malignancy: short axis (>1 cm), heterogeneous echogenicity, absence of CHS, presence of CNS, and blue-dominant elastographic images.

### 3.3. Diagnostic Test Parameters of Each EBUS Feature and Predictive Scoring System of MLNs

Table 4 summarizes the diagnostic test parameters for each EBUS feature and each predictive scoring system of MLNs. To predict malignancy, heterogeneous echogenicity and absence of CHS had high sensitivity (93.1% and 89.1%, respectively), CNS and blue-dominant elastographic images had high specificity (97.7% and 89.0%, respectively), and heterogeneous echogenicity showed the highest diagnostic accuracy (87.2%).

EBUS features with *p*-values < 0.05 in the multivariable analysis (Table 3) were included in the novel predictive scoring system (Figure 1). We scored from 0 to 5 points according to the number of EBUS features suggesting MLNs (short axis (>1 cm), heterogeneous echogenicity, absence of CHS, presence of CNS, and blue-dominant elastographic image). Figure 3a shows the ROC curve for the scoring system. The area under the curve (AUC) was 0.894 (95% confidence interval (CI): 0.868–0.920), and the cut-off score was 3 (sensitivity, 77.9%; specificity, 91.8%). Compared with this novel scoring system, the AUCs of the scoring system with all eight EBUS features, only six B-mode features, and the Canada LN score [11] (Figure 1) were lower at 0.857 (95% CI: 0.826–0.889, *p* < 0.001), 0.840 (95% CI; 0.807–0.873, *p* < 0.001), and 0.756 (95% CI: 0.719–0.792, *p* < 0.001), respectively (Figure 3b–d).

## 4. Discussion

In this study, we developed a novel scoring system by evaluating the diagnostic accuracy of eight EBUS features in B-, power/color Doppler, and elastography modes because only few studies have attempted to differentiate between benign LNs and MLNs using three EBUS modes simultaneously. The multivariable analysis showed that the B-mode features of the short axis (>1 cm), heterogeneous echogenicity, absence of CHS, and presence of CNS, as well as blue-dominant elastographic images, were independent predictive factors of MLNs. Our novel scoring system combining those EBUS features would predict MLNs more accurately than all eight EBUS features, only the six B-mode features, or the Canada LN score [11].

Numerous studies have reported the diagnostic yields of each EBUS B-mode feature in MLNs [13,14,16,26,27,28]. The diagnostic test parameters of the EBUS B-mode features differed in each study; however, the high sensitivity of the absence of CHS and high specificity of the presence of CNS and blue-dominant elastographic images were consistent with the findings of previous reports [20,26,27,29,30]. Our results showed that heterogeneous echogenicity had the highest diagnostic accuracy (87.2%) compared with the remaining EBUS features. Some studies have reported similar tendencies (82.5–87.5%) [12,14], and we believe that the use of multi-EBUS features is involved in the high accuracy of heterogeneous echogenicity noted in this study. LNs were simultaneously assessed using the B-, power/color Doppler, and elastography modes. The presence or absence of blood flow and differences in stiffness within LNs are informative findings for evaluating echogenicity. These findings of other modes may have affected the evaluation of echogenicity and led to more accurate and sensitive results.

One difference between this study and previous reports is the low sensitivity and accuracy of elastography. This may be due to the calculation method for diagnostic test parameters. In this study, we divided elastographic images into two groups (blue and others), although a relevant previous study divided the images into three groups and calculated the diagnostic test parameters after excluding the intermediate group [19]. Another reason may be the difference in the evaluation method (i.e., qualitative—which we adopted for this study—or quantitative). One prospective study reported no difference between the accuracy of qualitative and quantitative elastographic evaluation [21]; on the other hand, Verhoeven et al. compared four evaluation modalities—two qualitative scoring systems and two quantitative methods (strain histogram and strain ratio)—and reported that quantitative elastography evaluations using the strain histogram technique was the most accurate method of predicting MLNs [31]. In fact, the accuracy of elastography in this study was clearly lower than that reported for quantitative elastographic evaluation using the strain histogram method in a prospective multicenter study [32]. This may be due to the fact that image pattern diagnosis (qualitative evaluation) is less objective and is an operator-dependent evaluation method. The strain histogram method, which involves quantifying the relative strain found in manually selected LN areas and presenting it as a histogram of strain counts, is considered an objective, operator-independent method [31]. Although this study had a retrospective design and only image pattern diagnosis could be used in the evaluation, a more optimal evaluation of elastographic images may help develop a more accurate scoring system. However, the qualitative elastographic evaluation used in this study is simpler and faster than other methods and would be easier to perform during sedated EBUS-TBNA, when the available time is limited.

The ultrasound elastography currently available for EBUS is strain elastography, the technique used in this study, which measures distortion that is negatively correlated with tissue stiffness. The distribution of this distortion is a relative index that can vary with the degree of tissue compression [33,34]. On the other hand, strain wave elastography, which is also used for respiratory diseases such as superficial LNs, pleura, and lung lesions [35,36], measures shear wave velocity, which is positively correlated with tissue hardness. Since ultrasonic waves propagate faster in hard tissues, tissue hardness can be measured by measuring the propagation speed of shear waves generated in the region of interest [33,34]. Shear wave elastography can be used to measure the absolute values of elastic modulus, and if this method becomes available for EBUS, the evaluation of LNs by elastography may become more accurate.

Another difference is that vascular patterns were not significant independent predictors of MLNs in our study. Nakajima et al. reported that 87.5% of LNs with grade 2/3 vascular patterns and/or bronchial artery inflow signs were malignant, while 84.7% of those without both were benign [17]. This discrepancy may be caused by the difference in study design. Nakajima et al. retrospectively collected the data, whereas we evaluated the images during the EBUS-TBNA procedure. Another prospective study also revealed low sensitivity (51.0%) and specificity (45.3%) in MLN prediction [12]. LN assessment with the power/color Doppler mode requires flow gain adjustments for each LN. However, assessments during the procedure may lack precision as quick decisions are required. Notably, because we did not evaluate the bronchial artery inflow sign in this study, its influence was unknown.

This study also suggested a novel simple scoring system, which revealed the highest predictive performance among all scoring systems [11,12,13,14,26,27,28]. Three retrospective studies reported that B-mode features combined with elastography could predict MLNs more accurately than B-mode alone [20,29,30]. To the best of our knowledge, no study has prospectively evaluated LNs using the three EBUS modes. The results of the multivariable analysis indicated that the elastography mode should be included in the scoring system to predict MLNs. Recently, B-mode EBUS image analysis using artificial intelligence has been reported [15], although the diagnostic accuracy of MLNs was only 72.9%. We believe that using multi-EBUS features would potentially improve insufficient diagnostic test parameters.

This study has some limitations. First, it was conducted at a single center. In addition, the same bronchoscopist did not perform all EBUS-TBNA procedures; therefore, the operator’s skills may have somewhat affected the detection of EBUS features. Moreover, this study was only a derivation cohort study to construct a novel scoring system and did not include a validation cohort study. However, all 594 LNs were prospectively evaluated during the procedure by two or more bronchoscopists with sufficient years of experience, which is considered to be representative of real-world data. Furthermore, this study included patients from various backgrounds. Non-solid tumors, such as lymphoma, usually show different EBUS patterns [16]. Therefore, they may have different results; however, only seven non-solid tumor cases were included in this study. In addition, the pathological results were revealed only after the procedure; therefore, we decided to include them in the analyses. Further multi-center trials are required to confirm the results of this study.

## 5. Conclusions

Evaluation of LNs with multi-EBUS features would predict MLNs more accurately than B-mode features alone. Our novel scoring system using five EBUS features (short axis (>1 cm), heterogeneous echogenicity, absence of CHS, presence of CNS, and blue-dominant elastographic images) is useful for predicting MLNs during procedures.

## Figures and Tables

**Figure 1 cancers-14-05355-f001:**
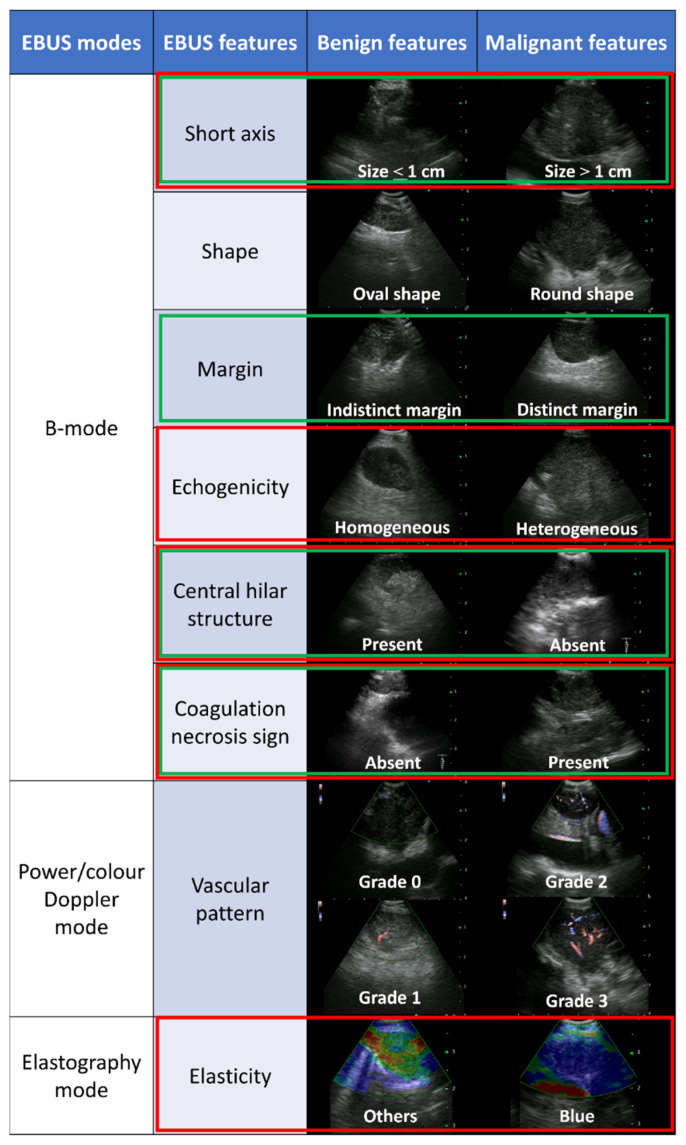
Representative endobronchial ultrasound (EBUS) features suggestive of benign and malignant lymph nodes in each EBUS mode and EBUS features included in each scoring system. In total, eight EBUS features were evaluated; four features surrounded by the green square show the Canada lymph node score, and five features surrounded by the red square show the novel scoring system.

**Figure 2 cancers-14-05355-f002:**
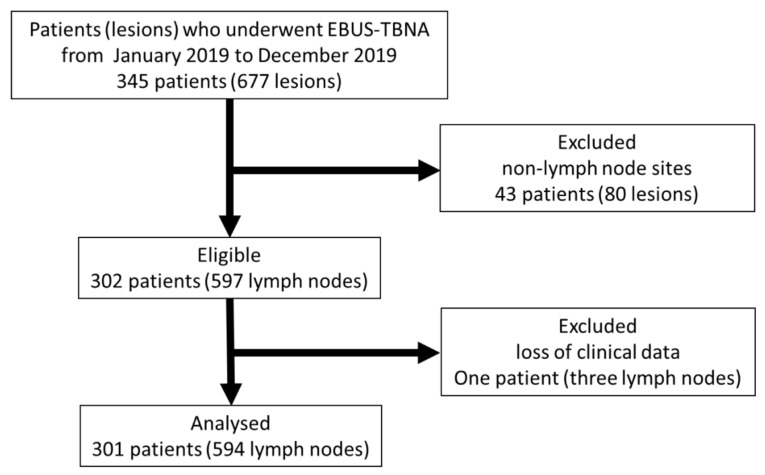
Flow chart of patient and lymph node recruitment. EBUS-TBNA, endobronchial ultrasound-guided transbronchial needle aspiration.

**Figure 3 cancers-14-05355-f003:**
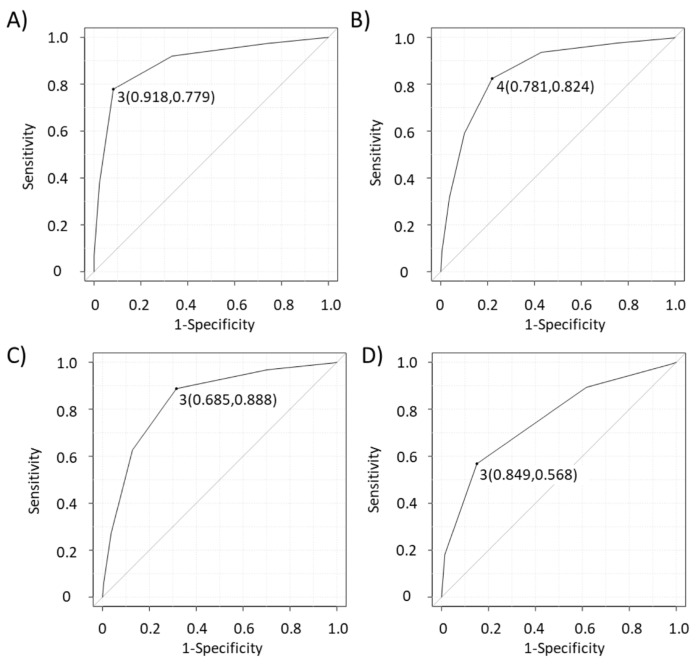
Discriminatory power for the prediction of malignant lymph node with novel scoring system (**A**), all eight EBUS features (B-, power/color Doppler, and elastography mode) (**B**), only six B-mode EBUS features (**C**), and the Canada lymph node score (**D**). The cut-off values derived from the receiver operating curve are 3.0, 4.0, 3.0, and 3.0, respectively. Their sensitivity and specificity for predicting malignancy are 77.9% and 91.8%, 82.4% and 78.1%, 88.8% and 68.5%, and 56.8% and 84.9%, respectively. The areas under the curve are 0.894 (95% CI: 0.868–0.92), 0.857 (95% CI: 0.826–0.889), 0.840 (95% CI: 0.807–0.873), and 0.756 (95% CI: 0.719–0.792), respectively. EBUS, endobronchial ultrasound; CI, confidence interval.

**Table 1 cancers-14-05355-t001:** Baseline characteristics of patients.

Characteristics	Numbers of Patients (N = 301) (%)
Sex	
Male	199 (66.1)
Age, years	66 (16–91)
Smoking history	
Never	74 (24.6)
Past	143 (47.5)
Current	84 (27.9)
Final diagnosis per patient	
Malignant	271 (90.0)
Lung cancer	233 (77.4)
Adenocarcinoma	117 (38.9)
Squamous cell carcinoma	39 (13.0)
Neuroendocrine tumor	63 (21.0)
Other non-small cell carcinoma	14 (4.7)
Non-pulmonary malignancies	38 (12.6)
Breast cancer	14 (4.7)
Gastrointestinal cancer	10 (3.3)
Hematologic cancer	4 (1.3)
Other cancer	10 (3.3)
Benign	30 (10.0)
Sarcoidosis (definite or suspect)	15 (5.0)
Infection	2 (0.7)
Others	3 (1.0)
Non-specific finding	10 (3.3)

Data are presented as numbers or medians (ranges). Other cancer (number of patients); prostate cancer (2), renal cell carcinoma (2), thymic carcinoma (2), uterine sarcoma (1), laryngeal cancer (1), thyroid cancer (1), and ovarian cancer (1). Others (benign) (number of patients); goiter (1), pericardial cyst (1), and not clearly defined benign tumor (1).

**Table 2 cancers-14-05355-t002:** Baseline characteristics of lymph nodes.

Characteristics	Number of Lymph Nodes (N = 594) (%)
Lymph node location	
Subcarinal (7)	162 (27.3)
Lower paratracheal (4R, 4L)	177 (29.8)
Hilar (10R, 10L)	19 (3.2)
Interlobar (11s, 11i, 11L)	131 (22.1)
Others (2R, 2L, 3p, 5, 8, 9, 12, 13)	105 (17.7)
Short axis on CT, cm	
≥1	327 (55.1)
<1	267 (44.9)
Short axis size on CT, cm	1.1 (0.3–5.6)
SUVmax on PET	
≥2.5	359 (60.4)
<2.5	127 (21.4)
Not evaluated	108 (18.2)
Final diagnosis per lymph node	
Malignant	375 (63.1)
Benign	219 (36.9)

Data are presented as numbers or medians (ranges). CT, computed tomography; PET, positron emission tomography; SUVmax, maximum standardized uptake value.

**Table 3 cancers-14-05355-t003:** Diagnostic yields of each EBUS feature for malignant lymph nodes.

EBUS Modes and Features	Numbers of Malignant LNs/Total LNs (%)	Univariable *p*-Value ^†^	Odds Ratio (95% CI) ^‡^	Multivariable *p*-Value ^‡^
B-mode				
Short axis		<0.001	1.860 (1.090–3.150)	<0.001
≤1 cm	130/275 (47.3)			
>1 cm	245/319 (76.8)			
Shape		0.104	0.970 (0.518–1.820)	0.925
Oval	283/461 (61.4)			
Round	92/133 (69.2)			
Margin		<0.001	0.561 (0.249–1.260)	0.163
Indistinct	68/85 (80.0)			
Distinct	307/509 (60.3)			
Echogenicity		<0.001	20.40 (11.30–36.50)	<0.001
Homogeneous	26/195 (13.3)			
Heterogeneous	349/399 (87.5)			
CHS		<0.001	1.910 (1.020–3.560)	0.043
Presence	41/156 (26.3)			
Absence	334/438 (76.3)			
CNS		<0.001	3.860 (1.370–10.90)	0.011
Absence	273/487 (56.1)			
Presence	102/107 (95.3)			
Power/color Doppler mode				
Vascular pattern		<0.001	1.060 (0.613–1.840)	0.827
Grades 0–1	172/334 (51.5)			
Grades 2–3	203/260 (78.1)			
Elastography mode				
Elasticity		<0.001	3.460 (1.830–6.560)	<0.001
Others	236/431 (54.8)			
Blue dominant	139/163 (85.3)			

Data are presented as numbers. ^†^ Calculated using Fisher’s exact test. ^‡^ Calculated using logistic regression analysis. EBUS, endobronchial ultrasound; LNs, lymph nodes; CHS, central hilar structure; CNS, coagulation necrosis sign; CI, confidence interval.

**Table 4 cancers-14-05355-t004:** Diagnostic test parameters of each EBUS feature and predictive scoring system for malignant lymph nodes.

**EBUS Features and Scoring Systems**	**Sensitivity (%)**	**Specificity (%)**	**PPV (%)**	**NPV (%)**	**Accuracy (%)**
B-mode					
Short axis (>1 cm)	65.3	66.2	76.8	52.7	65.7
Shape (round)	24.5	81.3	69.2	38.6	45.5
Margin (distinct)	81.9	7.8	60.3	20.0	54.5
Echogenicity (heterogeneous)	93.1	77.2	87.5	86.7	87.2
CHS (absence)	89.1	52.5	76.3	73.7	75.6
CNS (presence)	27.2	97.7	95.3	44.2	53.2
Power/color Doppler mode					
Vascular pattern (Grades 2–3)	54.1	74.0	78.1	48.5	61.4
Elastography mode					
Elasticity (blue)	37.1	89.0	85.3	45.2	56.2
Scoring system					
Novel scoring system	77.9	91.8	94.2	70.8	83.0
Eight EBUS features	82.4	78.1	86.6	72.2	81.0
Six B-mode features	88.8	68.5	82.8	78.1	81.3
Canada lymph node score	56.8	84.9	86.6	53.4	67.2

Data are presented as percentages. EBUS, endobronchial ultrasound; PPV, positive predictive value; NPV, negative predictive value; CHS, central hilar structure; CNS, coagulation necrosis sign.

## Data Availability

The dataset supporting the conclusions of this study is presented within the article. A detailed clinical dataset is not available to protect the privacy and confidentiality of the research subjects.

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
