# Peer review of "Predicting Malignant Lymph Nodes Using a Novel Scoring System Based on Multi-Endobronchial Ultrasound Features"

_cancers, 2022, doi:10.3390/cancers14215355_

Round 1

Reviewer 1 Report

Endobronchial ultrasound (EBUS) is a very important tool for the diagnosis of lymph nodes. During this procedure, several characteristics can be studied to differentiate between malignant and benign nodes.

The present study analyzes a retrospectively court of patient who underwent EBUS, evaluating the EBUS features (B-, 16 power/color Doppler, and elastography modes) and establishing a novel scoring system.

I think that the argument is interesting, also considering the wide and growing use of EBUS.
The court of the study is enough numerous and data are well presented. The weakness and the limits of the (retrospective) study are correctly presented by the authors.

I consider this article suitable for pubblication.

Reviewer 2 Report

Review Manuscript ID: cancers-1962671

This is retrospective analysis of a large sample of nodes in order to determine a composite score for predicting malignancy based on imaging features of EBUS based ultrasound characterization.

I applaud the authors for their effort.

Yet, in this paper they failed to include and address knowledge arising from the only, large, prospective multicenter study on EBUS strain elastography trial (E-predict) and systematic evalution of the technical bounderies of elastography assessment in EBUS. I would like to urgently advice the authors to study the available open access literature on this topic and relate their findings to this prior knowledge [see: Respiration 2019;97:337–347 · 10.1159/000494143 and Respiration 2020;99:484–492 · 10.1159/000507592].

Methodology

Inclusion criterium for EBUS is not stated explicitly, only 15 cases of sarcoidosis were found. It seems that lung cancer is the inclusion criterium but this should be clearly stated. What whas the final prevalence of lung cancer in this population?

In the dataset the gold standard for benign nodes is determined as a 6 months follow up on imaging. How many of these 219 benign nodes had histological confirmation of benign disease by surgical evaluation, and how many only based on imaging. In view of the oncological inclusion, 6 months is a very short observation period.

How many nodes had representative samples of lymph nodes, how many were inconclusive? 

And how is your scoring system influenced by PET imaging. Did all patients have PET available at the time the EBUS was performed? PET will greatly influence the endoscopist directing the nodes to aspirate, regardless of all other features.

Minor

Line 72: elastography measures relative strain, it is not objective; it only color-codes the relative strain within a single direction.

Line 74: hardness (nor stiffness) is not the scientifically correct term to describe strain, which can be measured by elastography.

Reviewer 3 Report

Dear Authors,

I read the paper titled "Predicting malignant lymph nodes using a novel scoring system based on multi-endobronchial ultrasound features" with great interest.

The study is scientifically sound and interesting, representing innovation but also continuation of some earlier research.

It gives me great pleasure to accept this paper without even minor revision in its scientific part.

Congratulations and keep up the good work.

Kind regards

Reviewer

Round 2

Reviewer 2 Report

in reply to my previous review the authors have chosen to add the references I suggested but they did not address the technical aspects and interpretation  thereof, nor addressed the other issues.

I refer to the previous evaluation for more details 

Round 3

Reviewer 2 Report

thank you

no more comments